# Mitochondrial Dysfunction and Metabolic Reprogramming in Obesity and Asthma

**DOI:** 10.3390/ijms25052944

**Published:** 2024-03-03

**Authors:** Paige Hartsoe, Fernando Holguin, Hong Wei Chu

**Affiliations:** 1Department of Medicine, National Jewish Health, Denver, CO 80222, USA; 2Department of Medicine, University of Colorado Anschutz Medical Campus, Aurora, CO 80045, USA

**Keywords:** obesity, asthma, mitochondrial dysfunction, metabolic reprogramming, inflammation, reactive oxygen species, mesenchymal stem cells

## Abstract

Mitochondrial dysfunction and metabolic reprogramming have been extensively studied in many disorders ranging from cardiovascular to neurodegenerative disease. Obesity has previously been associated with mitochondrial fragmentation, dysregulated glycolysis, and oxidative phosphorylation, as well as increased reactive oxygen species production. Current treatments focus on reducing cellular stress to restore homeostasis through the use of antioxidants or alterations of mitochondrial dynamics. This review focuses on the role of mitochondrial dysfunction in obesity particularly for those suffering from asthma and examines mitochondrial transfer from mesenchymal stem cells to restore function as a potential therapy. Mitochondrial targeted therapy to restore healthy metabolism may provide a unique approach to alleviate dysregulation in individuals with this unique endotype.

## 1. Introduction

About 26 million Americans suffer from asthma [1], which is characterized by reversible airway obstruction, airway hyperresponsiveness (AHR), and inflammation [2]. Obesity (BMI ≥ 30 kg/m^2^) is seen in about 40% of the adult US population, and is a major risk factor for asthma. Obese (vs. lean) asthma represents a significant health issue as patients have poor quality of life, respond less to inhaled steroids, and experience more severe symptoms and frequent exacerbations [3]. How excessive body weight impacts asthma depends on the underlying phenotype/endotype. In those with early-onset disease, before puberty, obesity is predominantly associated with increased type 2 (T2) inflammation (e.g., IL-13, eosinophils) and greater morbidity [4]. In contrast, adult-onset obese asthma, particularly in females, is characterized by less T2 airway inflammation and reduced exhaled nitric oxide (NO) levels [5,6,7], but more type 1 inflammation (e.g., IFN-γ) [8,9]. Thus, obese asthma is a complicated syndrome in which metabolism and inflammatory responses are highly interfaced in a complex pathogenesis. Mitochondrial dysfunction, characterized by decreased ATP production, membrane potential, and expression of mitochondrial complexes I, III, and IV, but increased respiration and reactive oxygen species (ROS) [10], has been observed in inflamed airways of people with asthma [11,12,13]. We are just at the beginning of understanding the role of mitochondrial dysfunction in obesity and asthma. The aim of this review is to briefly summarize current understanding of mitochondrial dysfunction in obese asthma and potential new therapeutic approaches to restore mitochondrial function.

## 2. Role of the Mitochondria in Health

The mitochondria, often referred to as the ‘powerhouse of the cell’, drives cellular function and metabolism. It is a double-membraned organelle, where the oxidation of carbohydrates, glucose, and fatty acids obtained through the diet occurs, followed by the generation of adenosine 5′-triphosphate (ATP) to provide energy for the cell [14]. There are multiple metabolic pathways that produce cellular energy including glycolysis, oxidative phosphorylation, and fatty acid metabolism. Changes that disrupt or shift the balance between these processes can cause mitochondrial dysfunction and lead to detrimental downstream effects.

Glycolysis, an important precursor to many other metabolic pathways, occurs in the cellular cytoplasm. During this process, glucose, a six-carbon molecule obtained through the diet, is oxidized by a series of enzymatic reactions into two molecules of pyruvate. Glycolysis generates a net total of two molecules of ATP per glucose molecule. Pyruvate is either converted to lactate, or shuttled into the mitochondria for conversion to acetyl-CoA under aerobic conditions. Acetyl-CoA is the precursor of the tricarboxylic acid (TCA) cycle, which occurs in the mitochondrial matrix [15]. Once this molecule enters the TCA cycle, it undergoes an eight-step oxidization process, in which it is converted to CO_2_ and 12 molecules of ATP are generated. High-energy byproducts such as NADH and FADH_2_ are created during this step, which later facilitate the synthesis of ATP through oxidative phosphorylation (OXPHOS) [16]. OXPHOS utilizes a series of oxidation–reduction reactions to generate ATP in the mitochondria through the electron transport chain (ETC) and is oxygen-dependent. The ETC is embedded within the inner membrane of the mitochondria and has four protein complexes, which are responsible for the transfer of electrons from the NADH and FADH_2_ generated in prior processes to oxygen [17]. The final complex in the ETC, complex V or ATP synthase, generates ATP through the energy provided from the gradient. OXPHOS generates a total of 30–36 molecules of ATP per glucose, and is much more efficient than glycolysis [18] but is generally more time-consuming. Under stress, such as infection or damage, cells are able to switch from OXPHOS to glycolysis to favor rapid turnaround of ATP [19]. The metabolic pathway utilized is dependent upon the cell type and microenvironment, and the preferred pathway is constantly shifting to meet changing energy demands.

In addition to energy production, mitochondria also play a critical role in the regulation of the immune response. Through secondary messengers, such as ROS, mitochondria are able to activate T cells through the Calcium (Ca^2+^)/nuclear factor of an activated T cell (NFAT) pathway [20]. Mitochondria also play an important role in viral immunity through the mitochondrial antiviral signaling (MAVS) protein. Viral RNA can trigger the activation of MAVS, which induces a downstream interferon response necessary for viral clearance [21]. The role of mitochondria in the differentiation and activation of immune cells is an emerging concept. When pro-inflammatory immune cells are activated, they require higher levels of energy for cytokine secretion or migration. These cells will often utilize glycolysis to quickly meet the demand. This phenomenon is seen in M1 macrophages, commonly referred to as pro-inflammatory, which exhibit decreased TCA cycle activity and upregulated glycolysis [20].

Ca^2+^ signaling is critical throughout the body, with Ca^2+^ acting as a secondary messenger in multiple pathways. Mitochondria store a large portion of the Ca^2+^ in the body and play a key role in maintaining Ca^2+^ homeostasis. Excessive ROS production can alter the movement of Ca^2+^ through mitochondrial membranes, leading to cellular apoptosis. Ca^2+^ is also able to influence the production of ATP by altering dehydrogenase activity [22,23].

All of the above processes are essential to maintaining cellular homeostasis and functionality. Mitochondria are the driving source of energy and regulation of many signaling pathways in the body. Alterations to these molecular processes in this organelle can have devastating effects observed throughout the body, leading to the pathogenesis of multiple diseases. There are various etiologies behind this dysfunction and they manifest in different ways in the body.

## 3. Mitochondrial Dysfunction and Metabolic Reprogramming

Mitochondrial dysfunction is a broad term to describe a state where mitochondria fail to adapt sufficiently or appropriately under different pathological conditions [24]. It can be presented at the morphological, metabolic, and functional levels, such as mitochondrial fragmentation, an excessive glycolytic rate, and ROS production and less ATP production. Metabolic reprogramming is highly associated with mitochondrial dysfunction. Metabolic reprogramming is generally defined as a process where cells adapt to a new environment or niche such as infection by rewiring their metabolism [25,26]. Hallmarks of metabolic reprogramming include increased glycolysis, mitochondrial changes and TCA cycle rewiring, increased lipid metabolism, changes in amino acid metabolism, and others. Research into the role of metabolic reprogramming in obese asthma is still in the early stage although it has been extensively investigated in the field of cancer research.

## 4. Common Forms of Mitochondrial Dysfunction and Metabolic Reprogramming

Maintaining mitochondrial homeostasis is crucial, and mitochondrial dynamics allow these organelles to adapt and respond to various stimuli by altering their size, shape, or location within the cell. The generation of new mitochondria, or biogenesis, is a process critical to the maintenance of homeostasis. Cells are able to respond to changes in energy demands by regulating the transcription factors involved in biogenesis to increase or decrease the production of new mitochondria [27]. Biogenesis is balanced by the selective removal of damaged or dysfunctional mitochondria through autophagy, commonly referred to as mitophagy [28]. Hinderance of either of these processes can lead to an imbalance between healthy, functional mitochondria and damaged organelles. 

Fission and fusion processes, as shown in Figure 1, are key in mitochondrial dynamics, and alterations of these processes appear to cause significant downstream effects. Fusion involves the integration of new mitochondria generated through biogenesis to the existing network by joining their mitochondrial membranes. This fusion allows the mitochondria to exchange metabolites and improve function in mitochondria that may have been damaged. Fusion is controlled by GTPase proteins Mitofusin 1 (Mfn1), Mitofusin 2 (Mfn2), and Optic atrophy protein (OPA1) [29]. Loss of function of these proteins can lead to mitochondrial fragmentation and enhanced cell apoptosis. The overexpression of these genes has also been found to increase mitophagy and reduce OXPHOS in pancreatic adenocarcinoma cells, suggesting that there is a fine balance in the maintenance of functionality in these cells [30]. The mitochondrial fusion factor (Mff) has been shown to activate MAVS in a mammalian cell culture. The phosphorylation of Mff by AMP-activated protein kinase (AMPK) can occur during mitochondrial dysfunction and the antiviral response is repressed [21]. Fission allows for the separation of damaged and healthy mitochondria and enhances the efficacy of mitophagy. The main proteins that are associated with the fission process are Dynamin-related protein 1 (Drp1) and mitochondrial fission protein 1 (Fis1). Disruptions in mitochondrial fission have been shown to lead to inflammation, decreased ATP production, and fragmentation [31]. The interactions of fission and fusion processes can vary across the cell type and shift in response to cellular changes. However, their balance is critical and the excess or diminution of either process can lead to the development of various diseases.

Electron transfer during aerobic energy production in the electron transport chain of the mitochondria generates metabolic byproducts, or ROS, such as superoxide (O_2_^•−^) and hydrogen peroxide (H_2_O_2_) [32]. Cells contain natural antioxidant mechanisms to remove ROS, such as superoxide dismutase (SOD) or catalase [33]. However, stress and mitochondrial dysfunction can lead to excessive levels of ROS that are inefficiently removed. This buildup of ROS in mitochondria can be detrimental to metabolic intermediates and enzymes required for OXPHOS. ROS has also been shown to cause metabolic reprogramming of the cell to favor glycolysis through hypoxia-inducible factor-1 (HIF1α) [34]. mtDNA is highly susceptible to oxidative damage. Increased ROS production in the mitochondria, due to metabolic shifts, can affect the mtDNA responsible for the encoding of subunits in the ETC, further progressing dysfunction [35].

Mitochondria also play a role in intercellular signaling and are able to induce various chemokine pathways. Upon cellular damage, mitochondrial DNA (mtDNA) is released into the cytosol and is able to activate a multitude of signaling pathways. Mitochondria have a bacterial origin, so when mtDNA is released into the cytosol, it is able to act as a damage-associated molecular pattern (DAMP) and trigger pattern recognition receptors (PRRs), such as Toll-like receptor 9 (TLR9), to activate pro-inflammatory pathways such as nuclear factor-kappa B (NF-κB). mtDNA is also able to activate the cyclic GMP–AMP synthase (cGAS) pathway and stimulate the STING interferon response [36,37]. Plasma mtDNA levels could be indicative of mitochondrial dysfunction and serve as an important biomarker for further characterizing disease.

## 5. Mitochondrial Dysfunction in Obesity and Asthma

Asthma is a heterogenous chronic inflammatory syndrome of the airways characterized by AHR, mucus hypersecretion, and airway remodeling [38]. The inflammatory process is coupled with elevated levels of pro-inflammatory cytokines and immune cells. Asthma can be broken down into multiple endotypes, or the distinct pathophysiological mechanisms. Patients with severe asthma can typically be classified as either T2-low or -high, and these can further be split into a phenotypic group. The phenotype refers to the observable clinical characteristics that a patient presents with including atopy, late-onset disease, smoking history, and obesity.

Obesity is both a comorbidity and a risk factor for developing asthma. In the United States, nearly 40% of individuals with asthma are obese [39], and the prevalence of asthma in adults who are obese is higher than in lean counterparts (11.1% vs 7.1%) [40]. Obesity alone is commonly associated with increased levels of systemic inflammation, elevated ROS, and mitochondrial dysfunction [41]. Individuals with obesity-related non-T2 asthma present with severe symptoms, frequent exacerbations, and resistance to conventional treatments such as inhaled steroids. This asthma endotype is commonly associated with elevated neutrophilic inflammation and increased interleukin (IL)-6 and -17 levels. Further investigation into biomarkers associated with this subset of asthma is necessary.

The nature of mitochondrial dysfunction and metabolic reprogramming in obese asthma and underlying mechanisms remain largely unknown. Distinct bioenergetic differences have been found between lean and obese asthma and healthy controls. Airway epithelial cells from people who are obese and asthmatic show increased basal and maximal oxygen consumption rates (OCRs) [5]. Elevated basal glycolytic rates were observed in airway epithelial cells [5] and airway smooth muscle cells [42] compared to healthy controls. ATP production was also elevated in obese asthmatic cells compared to healthy controls [5]. Additionally, increased levels of asymmetric dimethyl arginine (ADMA) in plasma of patients who are obese and asthmatic were reported [43]. Motta et al. showed that there was a unique metabolic phenotype of patients with both obesity and asthma that was able to be differentiated from those with solely one of the two diseases when analyzing breath condensates using nuclear magnetic resonance [44].

Similarly, another group looked at exhaled breath condensates of patients with chronic obstructive pulmonary disorder (COPD), asthma, and asthma–COPD overlap syndrome (ACOS). They showed that in patients with disease, there is a higher level of mtDNA/nDNA compared to healthy controls [45]. They found that patients presenting with non-T2 asthma had the highest concentrations of oxidative stress markers, as well as a higher concentration of mtDNA/nDNA compared to those with T2 [46].

Adipose-tissue-associated inflammation is still not widely understood in the context of obese asthma. Increased adipose tissue correlates to elevated levels of pro-inflammatory cytokines circulating in the individual’s blood such as IL-6 and tumor necrosis factor (TNF)-α, creating chronic systemic inflammation. There is also a newly proposed macrophage phenotype that may differ in metabolism compared to classically activated macrophages, referred to as metabolically activated. This has been considered as a result of abundant availability of energy sources, such as a high-fat diet. Adipose macrophages derived from individuals who are obese were found to have an increased number of mitochondria as well as elevated glycolysis and OXPHOS activity [47]. 

A murine asthma model demonstrated mitochondrial changes in the airway epithelium that were associated with decreased ATP production and airway remodeling [28]. In another lung model of an allergic response, a marked increase in ROS and a reduction in biogenesis was found in the airway epithelium [48].

In patients who are obese, there was a marked decrease in mitochondrial biogenesis due to a lowered oxidative capacity [49]. A reduced expression of Mfn2 mRNA, which is critical for the conservation of mitochondrial morphology, was observed in Zucker obese rats and the skeletal muscle of human individuals who are obese. This decrease in Mfn2 expression could contribute to the metabolic reprogramming seen in individuals who are obese since fragmentation has been associated with an increase in mitophagy [50]. In these individuals who are obese, a significant reduction in the capacity of antioxidant mechanisms, such as superoxide dismutase, is observed in addition to lowered blood serum levels of vitamins A, E, and C [51].

Obesity and allergens have both been shown to cause mitochondrial fragmentation, which may favor glycolysis [48]. The mechanism behind this fragmentation could be increased ceramide production as a result of palmitate obtained through a high-fat diet [52].

Under severe stress, such as that caused by the obese asthmatic condition, the reversal from glycolysis back to OXPHOS is delayed and excessive levels of pro-inflammatory cytokines are secreted. This severe inflammatory state could lead to the depletion of cell nutrients and overwhelm the metabolic system, leading to a total cellular shutdown [53].

## 6. Mitochondrial Dysfunction in Other Diseases

Although it has not been thoroughly investigated in obese asthma, mitochondrial dysfunction has been relatively well studied in the pathology of other diseases. Particularly, tissues that have a high energy demand, such as the heart, kidneys, brain, and skeletal muscles, exhibit indications of mitochondrial dysfunction. Looking at advancements in these areas of research could provide insight into potential mechanisms leading to the dysfunction of mitochondria in individuals with obesity and asthma as well as therapeutic targets.

A study of obstructive sleep apnea analyzed oxidative stress in human plasma samples by looking at reactive oxygen metabolites. They found a correlation between these metabolites and the mtDNA/nuclear DNA (nDNA) ratio, suggesting that there could be a link between ROS and excessive mtDNA release in obstructive lung diseases [54].

In neurodegenerative (e.g., Alzheimer’s and Parkinson’s) diseases, the dysfunction of metabolic enzymes is associated with a decline in mitochondrial bioenergetics and increased H_2_O_2_ [55]. A decrease in mitochondrial complex 1 enzyme activity has long been associated with the pathogenesis of Parkinson’s disease and an increase in oxidative stress [56]. Dysregulated mitochondrial Ca^2+^, causing cell death, is associated with the pathogenesis of neurodegenerative disorders such as Alzheimer’s and Huntington’s disease. Drug therapy with mitochondrially targeted antioxidants is used to reduce oxidative stress and cell death [23]. The role of genetics in these diseases is debated. Genes found to be associated with Parkinson’s have been found to also affect certain properties of the mitochondria. It is unproven whether mutations to mtDNA caused by elevated ROS could contribute to the development of these diseases [57].

In cardiovascular tissues, such as the heart, there is a large energy requirement and disruptions to the function of the respiratory chain can be detrimental. Mitochondrial dysfunction can progress heart failure due to increased ROS and diminished structural integrity of the cardiac tissues [58]. The deletion of mitochondrial complex 1 led to a metabolic shift in macrophages, favoring the pro-inflammatory phenotype. Further, mice with mitochondrial complex 1 deletion suffered from elevated inflammation levels and cell death as well as impaired healing after myocardial infarction, suggesting the importance of functional ETC activity for an effective anti-inflammatory response [59]. The formation of ‘megamitochondria’ has been observed in cardiac pathologies, resulting from dysregulated fusion caused by the upregulation of fusion proteins Mfn1 and Mfn2. These mitochondria are not functional, and can be eliminated through mitophagy. Fission proteins, such as Dnp1, are upregulated under ischemic conditions, and can increase mitochondrial fragmentation levels. Altering these protein expression levels has been shown to have a protective effect on cardiac mitochondria [48]. Examining the regulation of proteins involved in fission and fusion could alter mitochondrial function in other disease pathologies.

In cancer, metabolic reprogramming of cells is a main area of interest. Cancer cells have been found to significantly increase the rate at which they uptake glucose and produce lactate, associated with rapid increased ATP production, a phenomenon known as the Warburg effect [60]. Researchers have focused on the inhibition of glycolysis as a potential therapeutic target to halt errant growth of these cells. Newer studies suggest that cancer cells are metabolically heterogenous and the inhibition of OXPHOS could better prevent tumor growth through the use of drugs such as Metformin [61].

## 7. Mechanisms Driving Mitochondrial Dysfunction

It has been generally proposed that multiple genetic and environmental factors as well as lifestyles (e.g., diets) may regulate mitochondrial function [62]. Air pollution, respiratory viral infections, and increased levels of saturated fatty acids have been associated with obese asthma [63,64,65,66,67]. Each of these factors or in combination may result in mitochondrial dysfunction and metabolic reprogramming. Mechanistically, the integrated stress response (ISR) has been considered as a common pathway to drive mitochondrial dysfunction [68]. Whether obese asthma risk factors or stress signals activate the ISR (e.g., activating transcription factor 4 [ATF4]), and subsequently induce a transcriptional program, leading to mitochondrial dysfunction and metabolic reprogramming, remains to be determined.

These factors may affect ATP and ROS production, mitochondrial biogenesis, mitophagy, and mtDNA release. Multiple studies suggest that bacterial and viral infections induce mitochondrial dysfunction to evade host defense mechanisms [19,69]. Viruses have been shown to alter mitochondrial dynamics to promote infection. Viruses can enhance fission to inhibit mitochondrial apoptosis, induce mitochondrial fragmentation to disrupt signaling, and inhibit the antiviral response, as well as alter mitophagy [70]. Particulate matter has also been shown to impair mitochondrial function [71,72]. Environmental toxins, such as those from pesticides, have been associated with the inhibition of complex 1 in the ETC and linked to the pathogenesis of diseases such as Parkinson’s [56]. Immunometabolism is a broad term, encompassing immune cellular function, metabolism, and their interactions [73], which has been suggested as a key consequence of mitochondrial dysfunction [74]. Whether immunometabolism serves as a mechanism to sustain mitochondrial dysfunction remains unclear.

## 8. Therapeutic Approaches to Restore Mitochondrial Functions in Diseases

Currently, there is a substantial hole in treatments available for obesity-related severe asthma. Many reviews have concluded that there is a consistent improvement in asthma-related symptoms with weight loss, either through lifestyle changes or surgical intervention [75]. An improved lung function, AHR, and response to inhaled steroids have been observed following weight loss in patients with obesity and asthma [76,77]. Post weight loss, no significant changes to biomarkers associated with eosinophilic inflammation were found. In patients who were obese with type 2 diabetes, mitochondrial function was restored post-bariatric surgery, as determined by a decrease in the mtDNA copy number [78]. The effect of weight loss on mtDNA release has not been examined in patients with obesity and asthma.

Presently, therapeutic studies are focused on targeting mitochondrial dysfunction to restore homeostasis. Exogenous antioxidants to target ROS have been an area of interest. A mitochondria-targeted antioxidant (MitoQ) has been used in Parkinson’s studies as a protective factor against mitochondrial damage [53]. Mito-TEMPO has also been shown to decrease airway fibrosis in asthma [79]. Studies on the effects of these antioxidants in obese asthma are lacking. However, there is an ongoing pilot clinical trial (NCT04026711) that is randomizing 40 adults who are obese with asthma to MitoQ. This study will determine if, by reducing mitochondrial oxidative stress and downstream inflammation, MitoQ lessens obesity-mediated airway reactivity [80].

There has not been large success with studies targeting mitochondrial dynamics to treat obesity and this could be a new area of interest. One study found that a synthetic sphingolipid was able to alter mitochondrial dynamics in mice fed with a high-fat diet (HFD) and restore healthy body weight [52]. Others are interested by the idea of enhancing mitochondrial biogenesis as a therapeutic. Enhancement in biogenesis with therapeutics such as resveratrol may mitigate asthma symptoms by reducing oxidative stress and improving mitochondrial efficiency [48]. Future research should examine the regulation of proteins involved in mitochondrial dynamics as a potential target.

## 9. Cell-Based Therapeutic Approach: Mesenchymal Stem Cells

Mesenchymal stem cells (MSCs) are multipotent and easily grown in a cell culture, have low immunogenicity, and secrete immunomodulating and anti-inflammatory molecules and proteins [81]. These cells are abundantly present in bone marrow and adipose tissue. Multiple clinical trials have used MSCs as a therapeutic in cardiovascular disease and have demonstrated improved function in the heart [82]. Studies have also examined the therapeutic effect in neurodegenerative disorders such as Parkinson’s and Alzheimer’s diseases due to their anti-inflammatory effects, ability to differentiate, and regulation of oxidative stress [82]. Recently, there has been great interest in the use of these cells as a therapeutic agent in chronic lung diseases such as COPD and asthma.

MSCs have successfully been used to attenuate complications in multiple murine models representing different asthma phenotypes. MSC-derived extracellular vesicles (EVs) were shown to reduce airway inflammation and lung remodeling in an OVA-induced allergic asthma model [83]. Another study found that in a type 2 innate lymphoid cell (ILC2)-dominant allergic airway model, MSC EVs were able to decrease eosinophils and neutrophils, mucus production, and IL-5 and IL-13 release [84]. Limited human clinical trials using intranasal and intravenous delivery of MSCs to treat asthma have been registered and aim to determine the safety and efficacy of this treatment [85].

MSCs have also been used to ameliorate obesity-related complications. In diet-induced obese mice treated with human MSCs, a decrease in body weight was observed. Previous murine models have shown a successful restoration of the metabolic profile with decreased insulin sensitivity and decreased blood glucose levels following treatment with MSCs or their conditioned media. The restoration of balanced adipokine levels has also been observed [86].

Mitochondrial transfer is a naturally occurring phenomenon where neighboring cells can donate mitochondria to nearby damaged cells to rescue function. This has been observed across a variety of cell types. Studies have demonstrated the successful intercellular transfer of mitochondria from adipocytes to macrophages as well as from bone marrow mesenchymal stem cells to alveolar macrophages (AMs) [14]. Following this transfer in the AMs, OXPHOS was increased. MSCs have previously been shown to donate mitochondria to neighboring damaged cells [87]. Macrophages cultured in the presence of MSCs demonstrated a decreased inflammatory response and an increase in phagocytic activity [88].

One of the main benefits of using MSCs is their low immunogenicity. There have been successful reports of allogenic MSCs successfully being used as therapeutic agents; however, autologous MSCs are less likely to stimulate a host response. When using autologous MSCs, the hostile host metabolome and potential damage to the cells is an important consideration. In an HFD murine model, the MSCs had critical changes to their mitophagy, leading to excessive levels of dysfunctional mitochondria and a loss of function of intercellular mitochondrial transport [89]. Increased dysfunction was also seen in cultured myeloid cells treated with interferon (IFN)-γ and LPS, suggesting that obesity and the related pro-inflammatory cytokines can inhibit the restoration of homeostasis through intercellular mitochondrial transfer [90]. One study found that using pharmacological intervention, the functional capabilities of autologous MSCs from subjects who are obese can be restored and these cells are made viable. Antioxidant molecule pyrroloquinoline quinone (PQQ) was used to lower ROS and enhance bioenergetics in obese-derived MSCs, which were then co-cultured with epithelial cells that were pretreated with rotenone (Rot) to induce mitochondrial dysfunction. Successful mitochondrial transfer was observed between the MSCs and epithelial cells, as well as an attenuation of ROS, cell death, and mitochondrial shape in the recipient cells [89]. Furthermore, in an allergic murine model, MSCs derived from obese mice were cultured and treated with PQQ before transplantation. Reduction in airway hyperresponsiveness and mucus secretion was observed in mice treated with the modulated cells but not in mice fed with PQQ alone or treated with non-altered MSCs [89], suggesting the restoration of MSC function with PQQ.

Another consideration is the extent to which the MSCs are able to exert their effects in vivo. One clinical trial found that post intravenous injection in human subjects, MSCs were localized in the lungs and were only viable for 24 h [91]. More research into the mechanisms behind the migration of MSCs in the lungs and their fate is necessary.

One of the greatest challenges to MSC therapy is the harsh environment that the cells will encounter after transplantation. It has been proposed to pre-condition MSCs prior to treatment in order to enhance survival. For example, culturing MSCs with different cytokines, clinical drugs, or physical factors can enhance viability and cellular function. Culture media can be tailored to fit the unique endotype of the patient to enhance treatment outcomes. In the context of obese asthma, pro-inflammatory cytokines, such as IL-6 and IFN-γ, or adipokines could be added to the culture media. Exposing the cells to hypoxic conditions can also be protective [92]. Chemical agents can also be used to enhance MSCs’ role in asthma treatment. Antioxidants can reduce oxidative stress. Optimal conditions to culture MSCs prior to treatment should be examined for enhancement in survival and efficacy.

Personalized or precision medicine is a new frontier that is being explored. The complex nature of obese asthma with multiple phenotypes/endotypes and unique metabolic profiles makes individualized treatments such as MSC therapy necessary. MSCs as a therapeutic has not yet been explored in this late-onset obese phenotype. The abilities to recover the function of obese-derived autologous MSCs and culture in pre-conditioned media to increase viability are promising discoveries. This could be a novel therapy, summarized in Figure 2, for individuals and alleviate symptoms where currently offered treatments are failing.

## 10. Final Remarks

There are many unanswered questions in basic, translational, and clinical research of mitochondrial dysfunction and metabolic reprogramming in the context of obese asthma. These questions include, but are not limited to, the following:(1)What are the unique features or biomarkers of mitochondrial dysfunction and metabolic reprogramming in obese asthma as compared to lean asthma and healthy subjects?(2)Do mitochondrial dysfunction and metabolic reprogramming profiles predict the clinical and immunological phenotypes or endotypes such as pulmonary function and inflammation?(3)How may research findings from profiling mitochondrial dysfunction and metabolic reprogramming guide precision medicine in obese asthma?(4)Can we develop or refine mitochondria-targeted therapies to restore mitochondrial and metabolic homeostasis?(5)Are physical and dietary approaches effective in restoring mitochondrial function and metabolism in patients with obese asthma?

We propose some ideas to address the above questions.
(1)Perform cross-sectional and longitudinal studies in large cohorts of subjects with obese asthma and controls to obtain biological and metabolic data reflecting mitochondrial dysfunction and metabolic reprogramming. Research data will be analyzed in the context of clinical and immunological phenotypes or endotypes.(2)Develop and validate blood or airway biomarkers using minimally invasive approaches such as nasal brushing or lavage to indicate the levels of mitochondrial dysfunction and metabolic reprogramming in obese asthma.(3)Develop patient-specific therapies to restore mitochondrial homeostasis and improve clinical outcomes. This precision medicine approach needs more robust cutting-edge research to define how an individual subject with obese asthma presents a unique profile of mitochondrial dysfunction and metabolic reprogramming.(4)It is unlikely that a single mitochondria-target therapy will be effective to improve airway function in subjects with obese asthma. A combinational therapy targeting several pathways such as ROS and glycolysis may be necessary to maximize the therapeutic efficiency. Additionally, the interactions of mitochondria-target therapy and common asthma medications such as bronchodilators and corticosteroids should be considered to increase the therapeutic potency, while reducing any potential side effects.(5)Any mitochondria-target therapy (e.g., use of MSCs) should be combined with increased physical activity and an improved diet to further reduce pathobiological effects of mitochondrial dysfunction and metabolic reprogramming in obese asthma.

## Figures and Tables

**Figure 1 ijms-25-02944-f001:**
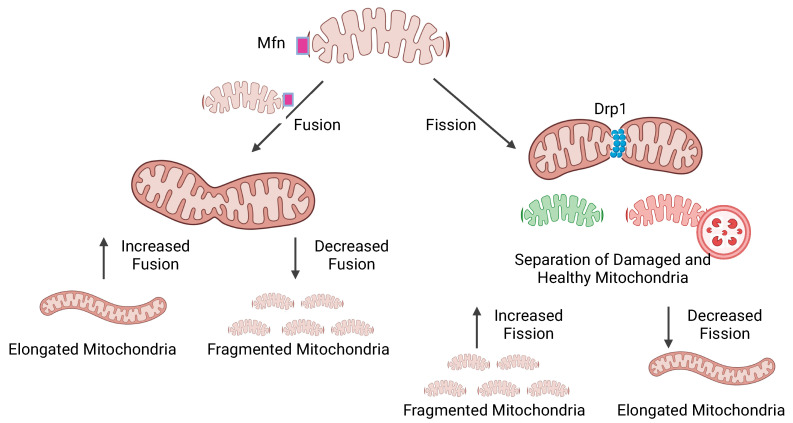
Mitochondrial dynamics play a key role in the regulation of their function. During fusion, two mitochondria are tethered together through the oligomerization of their Mfn proteins (shown in image in purple) and the new mitochondrion is incorporated into the existing network. Fission is the splitting of one mitochondrion into two separate organelles through the recruitment of Drp1 to allow damaged portions of mitochondria to more easily be removed through mitophagy. These processes are susceptible to changes in the cell’s environment and excessive changes can alter the mitochondrial integrity. Increased fusion and decreased fission are associated with mitochondrial elongation and resistance to apoptosis while on the opposite end of the spectrum, an increase in fission and decrease in fusion result in fragmented mitochondria, which are susceptible to apoptosis and mitophagy. Image created with BioRender.com.

**Figure 2 ijms-25-02944-f002:**
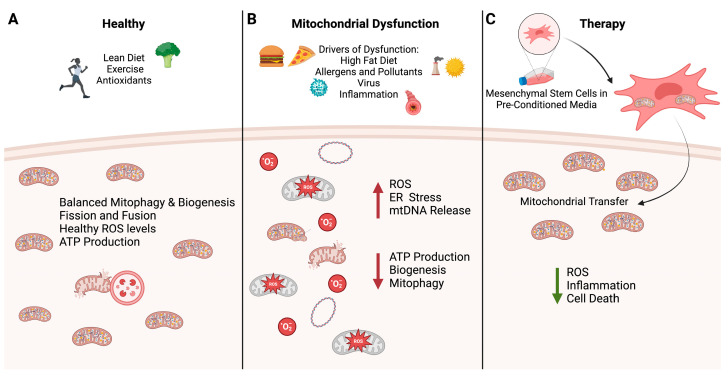
(**A**) Healthy cellular environment where metabolic function is regulated through lean diet and exercise. There is a balance of biogenetic processes and the successful removal of damaged mitochondria through mitophagy shown by a lysosome moving to the damaged mitochondria. (**B**) Factors such as a high−fat diet, environmental pollutants, allergens, and viral infections can lead to local and systemic inflammation, driving mitochondrial dysfunction. An increase in ROS and ER stress can cause excessive mtDNA release, and decreased mitophagy, ATP production, and biogenesis. (**C**) Host−derived mesenchymal stem cells can be cultured in media containing various treatments to pre-condition them for the harsh environment prior to transplantation. Once transplanted, mitochondrial transfer can occur, leading to a decrease in inflammation, ROS, and overall cell death as well as restoration of ATP production. Image created with BioRender.com.

## Data Availability

No new data created in this review.

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
