# Peer review of "Mitochondrial Dysfunction and Metabolic Reprogramming in Obesity and Asthma"

_ijms, 2024, doi:10.3390/ijms25052944_

Round 1

Reviewer 1 Report

Comments and Suggestions for Authors

·         Provide a detailed background on mitochondrial dysfunction and metabolic reprogramming in the context of obesity and asthma.

·         Please clearly discuss the potential mechanisms or pathways in the context of therapeutic Approaches to Restore Mitochondrial Functions in Diseases

·         Provide the future direction.

·         Does integrated stress response have an impact?

·         Add new keywords because all are included in the title.

Comments on the Quality of English Language

need to improve

Author Response

Please se

Point by Point Response to Reviewer #1

  1. Provide a detailed background on mitochondrial dysfunction and metabolic reprogramming in the context of obesity and asthma.

Thank you for this suggestion. We have further clearly defined the concepts of mitochondrial dysfunction and metabolic reprogramming and updated the current knowledge relating to obese asthma within the best of our ability, as this is still a relatively new concept. Lines 106-117.

“Mitochondrial dysfunction is a broad term to describe a state where mitochondria fail to adapt sufficiently or appropriately under different pathological conditions [25]. It can be presented at the morphological, metabolic and functional levels, such as mitochondrial fragmentation, excessive glycolytic rate and reactive oxygen species (ROS) production and less ATP production. Metabolic reprogramming is highly associated with mitochondrial dysfunction. Metabolic reprogramming is generally defined as a process where cells adapt to a new environment or niche such as infection by rewiring their metabolism [26,27]. Hallmarks of metabolic reprogramming include increased glycolysis, mitochondrial changes and tricarboxylic acid (TCA) cycle rewiring, increased lipid metabolism, changes in amino acid metabolism and others. Research into the role of metabolic reprogramming in obese asthma is still in the early stage although it has been extensively investigated in the field of cancer research.”

  1. Please clearly discuss the potential mechanisms or pathways in the context of therapeutic Approaches to Restore Mitochondrial Functions in Diseases

We think that this is an excellent suggestion. We feel that MSC therapy will exert its effects through a multitude of pathways and have expanded upon this section as well including multiple preclinical studies. (Lines 354-367)

“MSCs have successfully been used to attenuate complications in multiple murine models representing different asthma phenotypes. MSC derived extracellular vesicles (EVs) were shown to reduce airway inflammation and lung remodeling in a OVA induced allergic asthma model [83]. Another study found that in type 2 innate lymphoid cell (ILC2) dominant allergic airway model MSC EVs were able to decrease eosinophils and neu-trophils, mucus production, and IL-5 and IL-13 release [84]. Limited human clinical trials using intranasal and intravenous delivery of MSCs to treat asthma have been registered and aim to determine the safety and efficacy of this treatment [85].

MSCs have also been used to ameliorate obesity-related complications. In diet in-duced obese mice treated with human MSCs, a decrease in body weight was observed. Previous murine models have shown successful restoration of the metabolic profile with decreased insulin sensitivity and decreased blood glucose levels following treatment with MSCs or their conditioned media. Restoration of balanced adipokine levels has also been observed [86].”

Provide the future direction.

We agree that the final remarks lacked in terms of future directions. We have updated the section to contain bullets outlining the remaining questions in this specific field of research as well as 5 of our proposed future directions in lines 433-470.

“We propose some ideas to address above questions.

  • Perform cross-sectional and longitudinal studies in large cohorts of obese asthma subjects and controls to obtain biological and metabolic data reflecting mitochondrial dysfunction and metabolic reprogramming. Research data will be analyzed in the context of clinical and immunological phenotypes or endotypes.
  • Develop and validate blood or airway biomarkers using minimally invasive approaches such as nasal brushing or lavage to indicate the levels of mitochondrial dysfunction and metabolic reprogramming in obese asthma.
  • Develop patient-specific therapies to restore mitochondrial homeostasis and improve clinical outcomes. This precision medicine approach needs more robust cutting-edge research to define how individual obese asthma subject presents a unique profile of mitochondrial dysfunction and metabolic reprogramming.
  • It is unlikely that a single mitochondria-target therapy will be effective to improve airway function in obese asthma subjects. A combinational therapy targeting several pathways such as ROS and glycolysis may be necessary to maximize the therapeutic efficiency. Additionally, the interactions of mitochondria-target therapy and common asthma medications such as bronchodilators and corticosteroids should be considered to increase the therapeutic potency, while reduce any potential side effects.
  • Any mitochondria-target therapy (e.g., use of MSCs) should be combined with increased physical activity and improved diet to further reduce pathobiological effects of mitochondrial dysfunction and metabolic reprogramming in obese asthma.”

  1. Does integrated stress response have an impact?

We think that this is an excellent point that was raised. We agree that IRS is a key mechanism to mitochondrial dysfunction, however there is currently little to no knowledge on this topic specifically relating to obese asthma. Nonetheless, we have put a paragraph to introduce the concept in lines 296-302.

“Each of these factors or in combination may result in mitochondrial dysfunction and metabolic reprogramming. Mechanistically, the integrated stress response (ISR) has been considered as a common pathway to drive mitochondrial dysfunction (13). Whether obese asthma risk factors or stress signals activate the ISR (e.g., activating transcription factor 4 [ATF4]), and subsequently induce a transcriptional program leading to mitochondrial dysfunction and metabolic reprogramming remains to be determined.”

  1. Add new keywords because all are included in the title.

We appreciate the feedback. We have included some words from the title specifically, but  have removed Mitochondrial Transfer and added Inflammation and Reactive Oxygen Species to the keywords,

  “Keywords: Obesity; Asthma; Mitochondrial Dysfunction; Metabolic Reprogramming; Inflammation; Reactive Oxygen Species; Mesenchymal Stem Cells”

e the attachment

Reviewer 2 Report

Comments and Suggestions for Authors

Hartsoe et al. reviewed the importance of mitochondria health in obese asthma patients and the possibility of metabolic reprogramming through MSC therapy. It would be better to address some concerns for scientific sound before acceptance of the manuscript.

 1. Please emphasize the clinical importance of this study and strengthen the logical connection so that the readers can focus on the core topic.

 2. The general content about mitochondria is too long, so please reduce the content a little and strengthen the core topic, MSC therapy.

 3. Please provide possible grounds for how MSC therapy affects asthma endotypes.

 4. Please describe the abbreviation only once in the beginning and then only with the abbreviation (e.g., twice a full term and abbreviations, AHR on pages 1 and 6)

 5. Please describe the abbreviation for NFAT.

Comments on the Quality of English Language

Overall, it is well described in good English, and it is recommended that minor English expressions be refined once more.

Author Response

Point by Point Response to Reviewer #2

  1. Please emphasize the clinical importance of this study and strengthen the logical connection so that the readers can focus on the core topic.

We have added the suggested content pertaining to the clinical importance (lines 195-199). While it is still early in investigation, animal studies demonstrate the promising effect of MSC treatment to reduce asthmatic airway inflammation in different asthma models, as well as obesity related complications although the effect in the specific obese asthma phenotype has not been studied. (Lines 354-367)

“Individuals with obesity related non-T2 asthma present with severe symptoms, frequent exacerbations, and resistance to conventional treatments such as inhaled steroids. This asthma endotype is commonly associated with elevated neutrophilic inflammation and increased interleukin (IL)-6 and 17 levels. Further investigation into biomarkers associated with this subset of asthma is necessary. “

“MSCs have successfully been used to attenuate complications in multiple murine models representing different asthma phenotypes. MSC derived extracellular vesicles (EVs) were shown to reduce airway inflammation and lung remodeling in a OVA induced allergic asthma model [83]. Another study found that in type 2 innate lymphoid cell (ILC2) dominant allergic airway model MSC EVs were able to decrease eosinophils and neu-trophils, mucus production, and IL-5 and IL-13 release [84]. Limited human clinical trials using intranasal and intravenous delivery of MSCs to treat asthma have been registered and aim to determine the safety and efficacy of this treatment [85].

MSCs have also been used to ameliorate obesity-related complications. In diet in-duced obese mice treated with human MSCs, a decrease in body weight was observed. Previous murine models have shown successful restoration of the metabolic profile with decreased insulin sensitivity and decreased blood glucose levels following treatment with MSCs or their conditioned media. Restoration of balanced adipokine levels has also been observed [86].”

  1. The general content about mitochondria is too long, so please reduce the content a little and strengthen the core topic, MSC therapy.

            We appreciate these suggestions. We have condensed the general knowledge, but feel that the remaining content is beneficial to the discussions later on in the paper. We have also expanded upon the core content of MSC therapy.

  1. Please provide possible grounds for how MSC therapy affects asthma endotypes.

We think this is an excellent suggestion. We have included a brief introduction to asthma phenotypes and endotypes (lines 186-190). We have then discussed the potential benefits of MSC to different endotypes (lines 354-361) however, there are limited studies on the severe obese ashtma phenotype.

“Asthma can be broken down into varying endotypes, or the distinct pathophysiological mechanisms. Severe asthma patients can be typically be classified as either T2 low or high, and these can further be split into a phenotypic group. The phenotype refers to the observable clinical characteristics that a patient presents with including atopy, late onset disease, smoking history, and obesity. Individuals with obesity related non-T2 asthma present with severe symptoms, frequent exacerbations, and resistance to conventional treatments. This asthma endotype is commonly associated with elevated neutrophilic inflammation and increased interleukin(IL)-6 and 17 levels. Further investigation into biomarkers associated with this subset of asthma is necessary.”

  1. Please describe the abbreviation only once in the beginning and then only with the abbreviation (e.g., twice a full term and abbreviations, AHR on pages 1 and 6)/ Please describe the abbreviation for NFAT.

            Thank you for pointing these errors out. AHR specifically was corrected in line 190. The definition of NFAT has been provided on line 85 .We also went through the paper to ensure that abbreviations were provided only upon the first location in the paper.

“nuclear factor of an activated T cell (NFAT) pathway”

Reviewer 3 Report

Comments and Suggestions for Authors

The text explores mitochondrial dysfunction in obese individuals with asthma, focusing on the potential use of mesenchymal stem cells (MSCs) to restore mitochondrial function. It describes the complexity of asthma in obese individuals, highlighting the unique metabolic shifts and proposing MSC therapy as a potential intervention. The review suggests that targeting mitochondria could restore metabolic balance, alleviate chronic inflammation and stress, and discusses the regulation of proteins involved in mitochondrial dynamics as a potential target. The text emphasizes the need for further research to understand the specific alterations in mitochondrial function associated with obesity and asthma, with a focus on identifying biomarkers and potential therapeutic strategies. In detail, the abstract is detailed enough for the reader to comprehend the article without having to read the entire manuscript. There is no discrepancy between the abstract and the manuscript. The discussion is well-written and articulated.

Below few observations: - To speed up the reading process of the manuscript, include a graphical abstract; - Personal opinion: there is no conclusion. Separate them into a paragraph for more clarity; - It would be beneficial to include relevant references to support and enhance the content. Here is a suggested bibliography for consideration: 10.1016/j.arbres.2020.04.024; 10.1088/1752-7155/10/2/026005; - Improve the resolution of all figures; - In addition, the quality of the writing could have been much better.

In light of that, I believe the article demonstrates high scientific value and is worth reading.

Comments on the Quality of English Language

The English language is generally correct and readable. However, there are instances where sentence structures are complex and could be simplified for greater clarity. Additionally, paying attention to grammatical and punctuation details would enhance overall readability. The quality of the English language is good, but there is room for improvement in sentence construction, especially in complex scientific explanations. A review for minor grammatical errors and ensuring consistent language use throughout the document would elevate the overall quality.

Reviewer 4 Report

Comments and Suggestions for Authors

This study delved into the mitochondrial dysfunction involved in the obese asthma.

I have minor comments.

1. Figures should be described in the proper position of the manuscript.

2. Weight reduction is an essential treatment strategy for obese asthma. I would recommend summarizing the role of weight reduction in mitochondrial dysfunction and obese asthma.  

Author Response

This study delved into the mitochondrial dysfunction involved in the obese asthma.

I have minor comments.

  1. Figures should be described in the proper position of the manuscript.

Thank you for this comment. We have moved the figures to a location where we feel they appropriately fit.

  1. Weight reduction is an essential treatment strategy for obese asthma. I would recommend summarizing the role of weight reduction in mitochondrial dysfunction and obese asthma.

We agree that weight reduction is a crucial treatment strategy for obese asthma and have touched on this topic, including a brief mention of bariatric surgery, under our section on existing therapies in lines 317-325.

“Currently there is a substantial hole in treatments available for obesity related severe asthma. Many reviews have concluded that there is a consistent improvement of asthma related symptoms with weight loss, either through lifestyle changes or surgical intervention [75]. Improved lung function, AHR, and response to inhaled steroids have been observed following weight loss in patients with obesity and asthma [76,77]. Post weight loss no significant changes to biomarkers associated with eosinophilic inflammation were found. In obese patients with type 2 diabetes, mitochondrial function was restored post-bariatric surgery, as determined by a decrease in mtDNA copy number [78]. The effect of weight loss on mtDNA release has not been examined in patients with obesity and asthma.”

Round 2

Reviewer 2 Report

Comments and Suggestions for Authors

The authors have complemented the issues for scientific soundness and further investigation. This paper is now considered acceptable for adequately addressing some concerns.